# Maternal Experience of Domestic Violence, Associations with Children’s Lipid Biomarkers at 10 Years: Findings from MINIMat Study in Rural Bangladesh

**DOI:** 10.3390/nu11040910

**Published:** 2019-04-23

**Authors:** Shirin Ziaei, Ruchira Tabassum Naved, Anisur Rahman, Rubhana Raqib, Eva-Charlotte Ekström

**Affiliations:** 1Department of Women’s and Children’s Health, Uppsala University, SE-751 85 Uppsala, Sweden; Lotta.Ekstrom@kbh.uu.se; 2International Center for Diarrheal Disease Research, Bangladesh (ICDDR,B), Dhaka 1212, Bangladesh; ruchira@icddrb.org (R.T.N.); arahman@icddrb.org (A.R.); rubhana@icddrb.org (R.R.)

**Keywords:** domestic violence, children, lipid biomarkers, Bangladesh

## Abstract

The consequences of maternal experience of Domestic Violence (DV) on their children’s cardio-metabolic risk factors are unclear. We aimed to assess if maternal exposure to any or a specific form of DV (i.e., physical, sexual, emotional and controlling behaviors) before and after childbirth was associated with their children’s lipid biomarkers at the age of 10 years. A current observational sub-study of a larger MINIMat trial included a cohort of 1167 mothers and their children. The conflict tactic scale was used to record women’s experience of lifetime DV before and after childbirth at week 30 of pregnancy and at a 10-year follow up, respectively. Five ml of fasting blood sample was collected from the children to evaluate their lipid profile. Children of women who experienced any DV before childbirth had lower Apo A (β_adj_ −0.04; 95% CI: −0.08, −0.01). Women who experienced physical DV both before and after childbirth had children with higher triglycerides (β_adj_ 0.07; 95% CI: 0.01, 0.14). Children whose mother experienced sexual DV before birth had lower Apo A (β_adj_ −0.05; 95% CI: −0.08, −0.01) and High Density Lipoprotein (HDL) (β_adj_ −0.05; 95% CI: −0.10, −0.01) as well as higher Low Density Lipoprotein (LDL) (β_adj_ 0.17; 95% CI: 0.05, 0.29) and LDL/HDL (β 0.24; 95% CI: 0.11, 0.38). However, levels of LDL (β_adj_ −0.17; 95% CI: −0.28, −0.06), LDL/HDL (β_adj_ −0.12; 95% CI: −0.25, −0.00) and cholesterol (β_adj_ −0.13; 95% CI: −0.25, −0.02) were lower among the children of mothers who experienced controlling behavior after childbirth. Results from the current study suggest that maternal experience of physical or sexual DV might negatively affect their children’s lipid profile at the age of 10 years.

## 1. Introduction

The global burden of non-communicable and in particularly cardio-metabolic diseases is rapidly increasing in low- and middle-income countries [1,2]. According to World Health Organization (WHO), each year around 17 million people die prematurely due to Non Communicable Diseases (NCD) and 87% of these deaths happen in low-resource settings [3].

Development of cardio-metabolic diseases starts as early as fetal life and childhood [4,5]. A growing body of evidence suggests that early life stress exposure, including maternal experience of psychological stress may have long-lasting programming effects on metabolic and endocrine systems of the offspring and children and make them more susceptible to NCDs [6,7,8]. Alteration of the hypothalamic-pituitary-adrenocortical (HPA) axis and normal regulation of stress hormones such as cortisol has been suggested as an underlying mechanism for these changes [8,9]. Individuals who were exposed to maternal psychological stress during pregnancy have been shown to have higher body mass index, an unfavorable lipid profile and insulin resistance in their adult life [10,11,12].

In addition to prenatal life, infancy and early childhood are periods of continuous growth and epigenetic plasticity [13,14]. Exposure to stressful life events in this period might also have a potential adverse programming effect on children and make them more susceptible to NCDs later in life. It has been suggested that childhood exposure to stressful life events including adverse parenting, physical and or sexual abuse, and socioeconomic disadvantage can negatively affect lipid and carbohydrate metabolism and contribute to development of metabolic disorders, among them central adiposity, dyslipidemia and diabetes mellitus [15,16,17,18]. 

Experience of toxic stress during early life might potentially stem from an adverse family environment, especially maternal exposure to domestic violence (DV). Maternal experience of DV might negatively affect child cardio-metabolic health by increasing women’s psychological stress during and after pregnancy or by increasing child stress directly. While negative effects of DV on maternal and child mental and physical health have been constantly reported [19,20,21,22,23], little is known about the consequences of maternal experience of different forms of DV (i.e., physical, sexual, emotional and controlling behaviors) on child cardio-metabolic risk factors particularly in low and middle income settings where other adverse experiences like poverty may compound the negative effects of DV. Further, it is not clear if maternal exposure to specific forms of DV before and after child-birth have different effects on child cardio-metabolic risk factors. Identifying factors that are associated with developmental progression of cardio-metabolic diseases in early life is necessary in order to develop effective preventive interventions. 

In Bangladesh, DV is highly prevalent and more than 54% of ever-married women reported lifetime experience of physical and or sexual violence by their intimate partner [24]. Additionally, the country is facing an increasing rate of NCDs and cardio-metabolic diseases [25,26]. The current study aimed to assess if maternal exposure to any or a specific form of DV before, after and both before and after childbirth is associated with their children’s cardio-metabolic risk factors and in particular lipid biomarkers at the age of 10 years in rural Bangladesh.

## 2. Materials and Methods

### 2.1. Study Design and Pupulation

The study was embedded in a larger study registered as MINIMat trial (Maternal and infant nutrition intervention reg#ISRCTN16581394). MINIMat is a population-based individually randomized food and micronutrient supplementation trial which also included breastfeeding counselling intervention. The details of the study design and procedure have been described elsewhere [27,28]. In brief, the trial was conducted in Matlab, a rural sub- district located in 57 km south east of Dhaka, capital of Bangladesh. Since 1966, an ongoing Health and Demographic Surveillance System (HDSS) has been running in the area by International Centre for Diarrhoeal Disease Research, Bangladesh (icddr,b). As part of HDSS activities, community health workers of icddr,b visit each household and collect their demographic and selected health information on a monthly basis. During November 2001 till October 2003, all women who were identified pregnant in the area were invited to visit icddr,b clinics and they were enrolled in the trial if they were less than 14 weeks pregnant, had no serious illness and gave informed consent. Enrolled pregnant women (*n* = 4436) were randomly allocated into 2 food supplementation and 3 micronutrient supplementation groups in a 2 by 3 factorial design. As a third intervention, from the third trimester, women were randomly assigned to either receive standard ongoing usual health messages or breastfeeding counselling. Recruited women were followed up monthly at home and during week 14, 19 and 30 of their pregnancy at icddr,b clinics. From 4436 pregnancies, 3267 singleton live births were registered. Children were followed up from birth up to 2 years, at 4.5 and 10 years. Children of women who were born during April 2002 till June 2003 were invited to participate in the 10-year follow up (*n* = 1461), among them 1247 children participated in blood sampling. The main reasons for loss were absence of the children or their caretakers during the household or clinical visit or that the child refused to give blood.

### 2.2. Data Collection and Measurements

#### 2.2.1. Maternal and Child Characteristics

During a household or clinical visit, at approximately week 8 of pregnancy, maternal characteristics including weight, height, pregnancy history, age and educational level (none, 1–5 years, and 6 years and above), were recorded by using pre-coded questionnaire. Besides, women’s socio economic status was calculated based on land ownership, dwelling characteristics and household ownership of durable (such as bed, radio, TV, electric fan) and other goods [29] and further divided into quintiles. Through a birth notification system, newborn characteristics including sex and anthropometric measurements were recorded within 72 h after delivery by trained health workers.

#### 2.2.2. Explanatory Variables

##### Women’s Lifetime Experience of DV before Child’s Birth

A shortened and modified version of conflict tactic scale was used to record women’s lifetime experience of DV [30]. The scale included behaviorally explicit questions. It was adapted for use in Bangladesh and modified to capture DV. Women were interviewed during their clinical visits at around week 30 of pregnancy by a team of trained paramedics. The women were asked about their lifetime experience of different forms of physical, sexual, emotional DV as well as controlling behavior by their intimate partner and /or a family member. Based on women’s answers to the questionnaire, the following binary categories were created: experience of lifetime physical DV (yes/no), sexual DV (yes/no), emotional DV (yes/no) and controlling behavior (yes/no). Women’s experience of any lifetime DV (yes/no) was defined by their experience of one or more than one act of DV until their 30th week of pregnancy.

##### Women’s Experience of DV after Child’s Birth

During the 10-year follow up, the same questionnaire was used to evaluate women’s experience of DV since childbirth. The interviews were conducted during the household visit and women were asked if they have experienced any specific behaviorally explicit physical, sexual, emotional act of DV as well as controlling behavior by their intimate partner and /or a family member after their children were born. Similarly, the following binary categories were created: experience of physical DV (yes/no), sexual DV (yes/no), emotional DV (yes/no) and controlling behavior (yes/no). Women’s experience of any DV (yes/no) was defined by their experience of one or more than one form of DV after child birth.

Further, in order to evaluate cumulative effect of DV, additional variables (yes/no) were created to identify women who experienced any or specific forms of DV “*both before and after child birth*”.

#### 2.2.3. Outcome Variables

##### Children’s Lipid Biomarkers at 10 Years

During the 10-year follow up, overnight fasting blood sample of 5 mL were collected from the children. The plasma was centrifuged and separated within 4 h after blood collection and stored at −70 °C. The samples were Further transferred on dry ice to Uppsala University, Sweden where Plasma levels of apolipoprotein A-1 (ApoA1), apolipoprotein B (ApoB), cholesterol, High Density Lipoprotein (HDL), low density lipoprotein (LDL) and triglycerides were measured by immunoturbidimetry using the Architect ci8200^®^ Analyzer (Abbott Diagnostics, Abbott Park, Illinois, USA).

### 2.3. Ethical Considerations

The study was conducted in accordance with Helsinki Declaration. Informed written consent was obtained from participating women for the original trial and from all parents of participating children in each step of child follow-up. The study followed WHO ethical and safety guidelines for research on DV [31]. Interviews regarding DV were conducted in private by trained female paramedics. During the first round of interview at week 30 of pregnancy, women who reported experience of physical or sexual DV or suicidal ideation were offered mental health counselling [32]. At the 10-year follow up all the women interviewed were provided with information regarding services related to violence. The original and follow up study was approved by the ethical review committee of icddr,b (#2000-025 and #12-022 respectively), the 10 years follow-up of the children was additionally reviewed by the regional ethical review board at Uppsala University (#2012-346).

### 2.4. Statistical Analyses

Descriptive characteristics of the women and their children are presented as mean and standard deviation (SD) for continuous and frequency and percentage for categorical variables. Normality of the bio-markers data was evaluated by visual examination of Q-Q plots and histograms. Plasma concentrations of triglycerides were skewed and thus were reported as median and range and transferred into natural logarithm (Ln) before statistical analyses. We used chi-square test for categorical and student’s t test/ANOVA for continuous variables in order to compare characteristics of the participating women and those who had missing data as well as to evaluate factors associated with maternal experience of DV and children’s level of bio-markers. General linear models were used to estimate associations between exposures and outcomes of interest. Maternal level of education (none, 1–5 years, and 6 years and above), SES (quintiles), age (years), food and micronutrient supplementation groups, child sex and duration of exclusive breastfeeding (days) were considered as potential confounders. Food and micronutrient supplementation groups, child sex and duration of exclusive breastfeeding were removed from final model as they were not associated with maternal experience of DV and child’s level of biomarkers or did not change the model parameters. Further, in order to indicate the magnitude of potential effect of maternal exposure to DV on the children’s biomarker, we have calculated the Z-scores from the mean and SD of biomarkers in the reference population (with no experience of DV) as follows: (mean biomarker level of subjects – mean biomarker level of the reference population) ÷ (SD of the reference population). Statistical analyses were performed with statistical software package IBM SPSS Statistics version 24 (IBM, SPSS, Armonk, NY, USA)

## 3. Results

From the 1247 children who provided blood samples, maternal experience of DV was not available for 75 children, further, five children were missing biomarkers values, resulting in 1167 children included in the final analyses (Figure 1). Women who had missing data on DV, were younger (23.90 ± 5.2 years Vs 26.53 ± 5.9 years) compared to those with complete data. No other differences between demographic and anthropometric characteristics of the women and their children were found between the ones with missing and complete data.

Descriptive characteristics of the study sample are presented in Table 1. Women were around 26 years of age when they were enrolled in the original study and on average were 150 cm in height. Almost one third were undernourished with BMI<18.5 and approximately one-third had no formal education. More than 56% of the women experienced DV before birth of their children. The percentage of the women who had experience of DV increased after childbirth with 66% of the women reporting experience of at least one form of DV after their children were born. At the 10-year follow-up the children had a mean weight of 23.97 kg (SD 4.2) and mean height of 129.53 cm (SD 6.3). Around 28% of the children were stunted (height-for-age Z-score < −2) and almost half were underweight (weight-for-age Z-score < −2). 

### Effects of Maternal Experience of DV before and after Childbirth on Their Children’s Lipid Biomarkers

Maternal experience of any DV: Children of women with experience of any lifetime DV before childbirth had lower Apo A (β_adj_ −0.04, CI: −0.08, −0.01; Z-score: −0.26) in comparison to women who did not experience any DV (Table 2).

Maternal experience of any physical DV: Women who experienced any physical DV *both before and after* child birth had children with higher level of triglycerides (β_adj_ 0.07; CI: 0.01, 0.14; Z-score: 0.23) (Table 3).

Maternal experience of any sexual DV: Children whose mother experienced any sexual DV before birth had lower level of Apo A (β_adj_ −0.05, CI: −0.08, −0.01; Z-score: −0.28) and HDL (β_adj_ −0.05, CI: −0.10, −0.01; Z-score: −0.23) as well as higher level of LDL (β_adj_ 0.17, CI: 0.05, 0.29; Z-score: 0.24) and LDL/HDL (β_adj_ 0.24, CI: 0.11, 0.38; Z-score: 0.35) (Table 4). 

No significant association was observed between maternal experience of emotional DV and their children’s level of lipid bio-markers (Appendix A, See the Appendix A).

Maternal experience of any controlling behavior: Children of women who experienced any controlling behavior after childbirth had better lipid profile, i.e., the level of LDL (β_adj_ −0.17, CI: −0.28, −0.06; Z-score: −0.26), LDL/HDL (β_adj_ −0.12, CI: −0.25, −0.00; Z-score: −0.18) and cholesterol (β_adj_ −0.13, CI: −0.25, −0.02; Z-score: −0.19) was lower among these children. However, maternal experience of controlling behavior *both before and after* childbirth had a negative impact on children’s level of Apo A (β_adj_ −0.03, CI: −0.06, −0.00; Z-score: −0.22) (Table 5).

## 4. Discussion

MINIMat trial was performed in a population where DV against women is highly prevalent. Findings from this longitudinal study suggest that maternal experience of DV both before and after child birth is associated with level of lipid biomarkers in their children. However, the direction and strength of these associations were different based on the type of DV and time of the exposure. Among all forms of DV maternal experience of sexual DV before childbirth and physical DV *both before and after* childbirth were associated with negative lipid profile of their children. In contrast, children of women who experienced controlling behavior after child birth had better lipid profile.

We found children of women who experienced any physical or sexual DV were more likely to have an unfavorable lipid profile. Similar to these findings, in a study conducted in the U.S, experience of family conflict was associated with dyslipidemia among children with a mean age of 11 [33]. The adverse impact of maternal experience of DV on lipid profile of their children might operate through several pathways. (1) Experience of DV might increase maternal stress during pregnancy. Fetal exposure to maternal stress can permanently alter HPA function in offspring and make them more susceptible to metabolic syndrome including dyslipidemia later in life [14,34]. (2) Maternal stress due to experience of DV might negatively affect their caring capacities and manifest as hostile parenting [35]. In a longitudinal study conducted in Finland maternal hostile child rearing attitude was associated with a higher level of triglycerides in their children [16]. (3) Exposure to family conflict such as DV can directly increase children’s level of stress [36]. There is growing evidence that experience of psychological stress can alter stress regulatory systems including sympathetic adrenal medullary system and HPA axis and make the children more susceptible to NCDs including cardiovascular diseases, metabolic syndrome and dyslipidemia [8]. (4) In the families, with presence of DV against the mother, the children might be victims of DV themselves [37]. Individuals who experienced childhood abuse have been shown to have higher risk of cardiovascular diseases and negative lipid profile later in life [38]. 

Among all forms of DV, women’s experience of sexual DV before childbirth had the most negative association with their children’s lipid biomarkers. Prevalence of sexual partner violence is relatively high in Bangladesh [39] and it has been reported that almost 30% of the women in this country explained that their first sexual experience was forced [40]. Experience of sexual DV can bring several physical and mental health consequences to the women including unwanted pregnancies, sexually transmitted diseases, urinary tract infections as well as high level of anxiety, depression and negative self-image [41,42]. All these factors can increase women’s mental and physical stress level during and after pregnancy and negatively affect their offspring. 

Unlike sexual DV, the negative association of physical DV against women with their children’s lipid biomarkers was observed only in women who experienced physical DV *both before and after* childbirth. Previous studies showed that physical DV is the most common form of family violence that child might witness [43,44]. Children who were exposed to family violence have demonstrated higher level of emotional difficulties including experiencing of fear, depression, powerlessness and post-traumatic stress disorders [45]. It is possible that the unfavorable effects of maternal experience of physical DV *both before and after* childbirth on their children’s biomarkers mediated through the combination of maternal stress and high level of emotional stress that children experience.

While women who experienced controlling behavior *both before and after* child birth had children with a lower level of Apo A, women’s experience of controlling behavior after childbirth was positively associated with their children’s lipid profile. This association was unexpected, given that controlling behavior limits women’s level of autonomy [46] and increases their psychological stress [47]. Possible pathways through which women’s experiences of controlling behavior affect their children’s biomarkers needs more investigation. A potential explanation can be that experience of controlling behavior might limit women’s mobility and thus women stayed more at home and spent more time with their children. 

The strength of the current study derives from its relatively large sample size and inclusion of different forms and timing of DV in the analyses. All data was collected following standard operating procedures and the data collectors went through proper refresher trainings throughout the study period. Since the study was imbedded in an intervention trial, additional analyses were performed controlling for food and micronutrients as well as breast feeding counselling interventions with no important changes in effect estimates. Information on the dietary intake of the children is missing in this study. The diet of the children might have been influenced by maternal caretaking practices and/or their own level of stress. Previous studies shown that maternal stress has been associated with negative feeding style and decreased consumption of fruits and vegetables by their children [48,49]. Further, psychological stress has been shown to change the dietary pattern of children toward an unhealthy style [50]. Thus children’s dietary intake could lie within the causal pathway between maternal exposure to DV and child level of lipid biomarkers and controlling for that would have introduced bias in our findings. We found slight differences between women with missing data and the ones that were included in the analyses, where those with incomplete data were younger. However, given the relatively small proportion of women who had missing data (6%), it is unlikely that our results have been subjected to selection bias. Women’s experience of lifetime DV might have been affected by recall bias. Previous research suggests that under reporting of DV is common in women [51]; we tried to reduce the risk of such bias by using standardized questionnaire with behaviorally explicit questions, taking interviews in private and guaranteeing confidentiality. Nonetheless, if under-reporting is the case; it might have diluted the observed associations in our study. We found great overlap between different forms of DV with more than 58% of the women reporting experience of multiple forms of abuse. The pattern of DV is usually complex. It often combines different forms of abuse rather than a single form [52]. Thus, evaluating the independent effect of each form of DV might be challenging. However, adjusting for different forms of DV, did not change the associations in this study. Further, there might be residual confounders and factors that are associated with experience of DV and children’s level of biomarkers that we have not measured. Finally, due to the observational nature of the study making causal inference is not possible. 

## 5. Conclusions

Findings from the present study suggest that maternal experience of physical or sexual DV might have negative effects on their children’s lipid profile at the age of 10 years. Previous studies indicated that plasma lipids are moderately stable from childhood into adulthood [53,54] and each 1% increase in serum lipids such as cholesterol and LDL elevates the risk of cardiovascular disease by 2–3% [55,56]. Considering high prevalence of DV in this population and globally, interventions targeting women’s experience of DV might have potential long term public health implications and reduce the risk of NCDs in children. 

## Figures and Tables

**Figure 1 nutrients-11-00910-f001:**
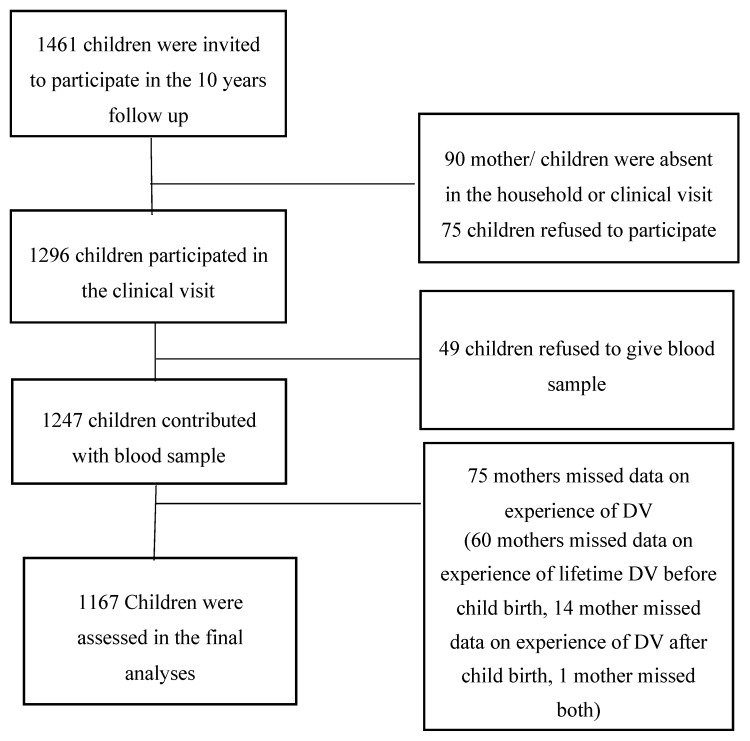
Fellow chart of participating children and their mothers.

**Table 1 nutrients-11-00910-t001:** Descriptive characteristics of participating children and their mothers.

Variable	*n* (%) or Mean ± SD (N = 1167)	Variable	*n* (%) or Mean ± SD (N = 1167)
Maternal characteristics at pregnancy week 8		Child characteristics	
Age in years	26.50 ± 5.9	Sex (female)	567 (48.8)
BMI	20.04 ± 2.6	Age in years	10.08 ± 0.1
Height	149.84 ± 5.2	Height (cm)	129.53 ± 6.3
Education		Weight (Kg)	23.97 ± 4.2
None	430 (36.8)	BMI (Kg/m^2^)	14.20 ± 1.6
1–5 years	264 (22.6)	Stunted	306 (28.2)
≥6 years	473 (40.5)	Underweight	414 (48.5)
Lifetime experience of DV before child birth			
Any DV	661 (56.6)	Level of lipid biomarkers	
Any physical DV	246 (21.1)	Apo A (g/l)	1.25 ± 0.18
Any sexual DV	312 (26.7)	Apo B (g/l)	0.74 ± 0.16
Any emotional DV	276 (23.7)	Apo B/Apo A	0.61 ± 0.16
Any controlling behavior	419 (35.9)	HDL (mmol/l)	1.11 ± 0.23
		LDL (mmol/l)	1.90 ± 0.67
Experience of DV after child birth		LDL/HDL	1.79 ± 0.73
Any DV	770 (66.0)	Cholesterol (mmol/l)	3.77 ± 0.70
Any physical DV	454 (38.9)	Triglycerides ^1^ (mmol/l)	0.95 (0.32–4.10)
Any sexual DV	439 (37.6)		
Any emotional DV	458 (39.2)		
Any controlling behavior	388 (33.2)		

^1^ The level of biomarker is presented as median and range. BMI, Body Mass Index; DV, Domestic violence; HDL, High Density Lipoprotein; LDL, Low Density Lipoprotein.

**Table 2 nutrients-11-00910-t002:** Association of children’s level of lipid biomarkers with maternal experience of any lifetime Domestic Violence (DV) before and after childbirth.

		Maternal Experience of Any Lifetime DV			
Biomarkers	Model	No Experience	Before Childbirth	After Childbirth	Both before and after Child Birth
			ß (95% CI)	ß (95% CI)	ß (95% CI)
Apo A (g/l)	Unadjusted	Ref	−0.04 (−0.08, −0.01) *	−0.02 (−0.06, 0.01)	−0.03 (−0.06, −0.00) *
	Adjusted ^1^	Ref	−0.04 (−0.08, −0.01) *	−0.02 (−0.05, 0.01)	−0.03 (−0.06, 0.00)
Apo B (g/l)	Unadjusted	Ref	0.00 (−0.03, 0.03)	−0.02 (−0.05, 0.01)	−0.00 (−0.03, 0.02)
	Adjusted ^1^	Ref	0.00 (−0.03, 0.03)	−0.02 (−0.05, 0.01)	−0.00 (−0.03, 0.02)
Apo B/Apo A	Unadjusted	Ref	0.02 (−0.02, 0.05)	−0.01 (−0.04, 0.02)	0.01 (−0.01, 0.04)
	Adjusted ^1^	Ref	0.02 (−0.02, 0.05)	−0.01 (−0.04, 0.02)	0.01 (−0.02, 0.03)
HDL (mmol/l)	Unadjusted	Ref	−0.04 (−0.09, 0.00)	−0.03 (−0.07, 0.01)	−0.03 (−0.07, 0.01)
	Adjusted ^1^	Ref	−0.04 (−0.09, 0.00)	−0.03 (−0.07, 0.01)	−0.03 (−0.06, 0.01)
LDL (mmol/l)	Unadjusted	Ref	−0.00 (−0.14, 0.13)	−0.00 (−0.12, 0.12)	0.03 (−0.08, 0.13)
	Adjusted ^1^	Ref	0.01 (−0.13, 0.14)	0.01 (−0.11, 0.13)	0.04 (−0.07, 0.15)
LDL/HDL	Unadjusted	Ref	0.08 (−0.07, 0.22)	0.03 (−0.10, 0.16)	0.09 (−0.03, 0.20)
	Adjusted ^1^	Ref	0.09 (−0.06, 0.23)	0.04 (−0.09, 0.17)	0.10 (−0.02, 0.21)
Cholesterol (mmol/l)	Unadjusted	Ref	−0.07 (−0.20, 0.07)	−0.10 (−0.22, 0.03)	−0.04 (−0.15, 0.07)
	Adjusted ^1^	Ref	−0.05 (−0.20, 0.09)	−0.08 (−0.20, 0.04)	−0.02 (−0.14, 0.09)
Triglycerides ^2^ (mmol/l)	Unadjusted	Ref	0.03 (−0.04, 0.10)	−0.02 (−0.08, 0.04)	0.01 (−0.04, 0.07)
	Adjusted ^1^	Ref	0.03 (−0.04, 0.10)	−0.02 (−0.08, 0.04)	0.01 (−0.05, 0.07)

* *P* < 0.05. ^1^ Models adjusted for maternal education, SES and age. ß and 95% of CI obtained using general linear models. ^2^ The level of biomarker was transformed using the natural logarithm, and effect estimate reported accordingly. DV, Domestic violence; HDL, High Density Lipoprotein; LDL, Low Density Lipoprotein; Ref, reference category.

**Table 3 nutrients-11-00910-t003:** Association of children’s level of lipid biomarkers with maternal experience of any physical DV before and after childbirth.

		Maternal Experience of Any Physical DV			
Biomarkers	Model	No Experience	Before Childbirth	After Childbirth	Both before and after Child Birth
			ß (95% CI)	ß (95% CI)	ß (95% CI)
Apo A (g/l)	Unadjusted	Ref	−0.03 (−0.07, 0.02)	−0.02 (−0.05, 0.00)	−0.03 (−0.06, 0.00)
	Adjusted ^1^	Ref	−0.02 (−0.07, 0.02)	−0.02 (−0.04, 0.01)	−0.02 (−0.05, 0.01)
Apo B (g/l)	Unadjusted	Ref	0.02 (−0.02, 0.06)	−0.01 (−0.03, 0.01)	0.01 (−0.02, 0.03)
	Adjusted ^1^	Ref	0.01 (−0.03, 0.05)	−0.01 (−0.04, 0.01)	0.00 (−0.03, 0.03)
Apo B/Apo A	Unadjusted	Ref	0.03 (−0.01, 0.06)	−0.00 (−0.02, 0.02)	0.02 (−0.01, 0.04)
	Adjusted ^1^	Ref	0.02 (−0.02, 0.06)	−0.00 (−0.03, 0.02)	0.01 (−0.02, 0.03)
HDL (mmol/l)	Unadjusted	Ref	−0.03 (−0.09, 0.02)	−0.03 (−0.06, 0.00)	−0.04 (−0.08, 0.00)
	Adjusted ^1^	Ref	−0.03 (−0.08, 0.03)	−0.02 (−0.05, 0.01)	−0.02 (−0.06, 0.02)
LDL (mmol/l)	Unadjusted	Ref	0.04 (−0.13, 0.20)	0.04 (−0.06, 0.13)	0.06 (−0.06, 0.17)
	Adjusted ^1^	Ref	0.05 (−0.12, 0.21)	0.05 (−0.05, 0.15)	0.07 (−0.05, 0.19)
LDL/HDL	Unadjusted	Ref	0.10 (−0.08, 0.28)	0.09 (−0.01, 0.19)	0.13 (−0.01, 0.25)
	Adjusted ^1^	Ref	0.10 (−0.08, 0.28)	0.09 (−0.02, 0.19)	0.12 (−0.01, 0.25)
Cholesterol (mmol/l)	Unadjusted	Ref	−0.01(−0.18, 0.16)	−0.06 (−0.15, 0.04)	0.01 (−0.11, 0.13)
	Adjusted ^1^	Ref	0.00 (−0.17, 0.18)	−0.04 (−0.14, 0.06)	0.03 (−0.10, 0.15)
Triglycerides ^2^ (mmol/l)	Unadjusted	Ref	−0.01 (−0.10, 0.07)	0.01 (−0.04, 0.06)	0.08 (0.02, 0.14) **
	Adjusted ^1^	Ref	−0.02 (−0.11, 0.07)	0.01 (−0.04, 0.06)	0.07 (0.01, 0.14) *

* *P* < 0.05, ** *P* < 0.01. ^1^ Models adjusted for maternal education, SES and age. ß and 95% of CI obtained using general linear models. ^2^ The level of biomarker was transformed using the natural logarithm, and effect estimate reported accordingly. DV, Domestic violence; HDL, High Density Lipoprotein; LDL, Low Density Lipoprotein; Ref, reference category.

**Table 4 nutrients-11-00910-t004:** Association of children’s level of lipid biomarkers with maternal experience of any sexual DV before and after childbirth.

		Maternal Experience of Any Sexual DV			
Biomarkers	Model	No Experience	Before Childbirth	After Childbirth	Both before and after Child Birth
			ß (95% CI)	ß (95% CI)	ß (95% CI)
Apo A (g/l)	Unadjusted	Ref	−0.05 (−0.08, −0.01) **	−0.01 (−0.03, 0.02)	−0.02 (−0.05, 0.02)
	Adjusted ^1^	Ref	−0.05 (−0.08, −0.01) **	−0.00 (−0.03, 0.02)	−0.02 (−0.05, 0.02)
Apo B (g/l)	Unadjusted	Ref	0.03 (−0.00, 0.06)	−0.01 (−0.03, 0.02)	0.02 (−0.01, 0.04)
	Adjusted ^1^	Ref	0.03 (0.00, 0.06) *	−0.01 (−0.03, 0.02)	0.02 (−0.01, 0.04)
Apo B/Apo A	Unadjusted	Ref	0.04 (0.01, 0.07)	−0.00 (−0.02, 0.02)	0.02 (−0.01, 0.05)
	Adjusted ^1^	Ref	0.04 (0.02, 0.07)	−0.00 (−0.02, 0.02)	0.02 (−0.01, 0.05)
HDL (mmol/l)	Unadjusted	Ref	−0.05 (−0.10, −0.01) *	−0.01 (−0.04, 0.03)	−0.00 (−0.04, 0.04)
	Adjusted ^1^	Ref	−0.05 (−0.10, −0.01) **	−0.00 (−0.03, 0.03)	−0.00 (−0.04, 0.04)
LDL (mmol/l)	Unadjusted	Ref	0.16 (0.04, 0.29) **	−0.03 (−0.12, 0.07)	0.04 (−0.08, 0.15)
	Adjusted ^1^	Ref	0.17 (0.05, 0.29) **	−0.02 (−0.12, 0.07)	0.04 (−0.08, 0.16)
LDL/HDL	Unadjusted	Ref	0.24 (0.11, 0.37) **	−0.01 (−0.11, 0.10)	0.04 (−0.09, 0.17)
	Adjusted ^1^	Ref	0.24 (0.11, 0.38) **	−0.01 (−0.11, 0.10)	0.04 (−0.09, 0.17)
Cholesterol (mmol/l)	Unadjusted	Ref	0.04 (−0.08, 0.17)	−0.04 (−0.14, 0.06)	0.08 (−0.04, 0.20)
	Adjusted ^1^	Ref	0.05 (−0.08, 0.17)	−0.03 (−0.14, 0.07)	0.08 (−0.04, 0.20)
Triglycerides ^2^ (mmol/l)	Unadjusted	Ref	0.06 (−0.01, 0.12)	−0.01 (−0.06, 0.04)	−0.01 (−0.07, 0.06)
	Adjusted ^1^	Ref	0.06 (−0.01, 0.12)	−0.02 (−0.07, 0.03)	−0.01 (−0.07, 0.05)

* *P* < 0.05, ** *P* < 0.01. ^1^ Models adjusted for maternal education, SES and age. ß and 95% of CI obtained using general linear models. ^2^ The level of biomarker was transformed using the natural logarithm, and effect estimate reported accordingly. DV, Domestic violence; HDL, High Density Lipoprotein; LDL, Low Density Lipoprotein; Ref, reference category.

**Table 5 nutrients-11-00910-t005:** Association of children’s level of lipid biomarkers with maternal experience of any controlling behavior before and after childbirth.

		Maternal Experience of Any Controlling Behavior			
Biomarkers	Model	No Experience	Before Childbirth	After Childbirth	Both before and after Child Birth
			ß (95% CI)	ß (95% CI)	ß (95% CI)
Apo A (g/l)	Unadjusted	Ref	−0.02 (−0.04, 0.01)	−0.01 (−0.04, 0.02)	−0.04 (−0.06, −0.00) *
	Adjusted ^1^	Ref	−0.02 (−0.04, 0.01)	−0.01 (−0.04, 0.02)	−0.03 (−0.06, −0.00) *
Apo B (g/l)	Unadjusted	Ref	−0.01 (−0.03, 0.02)	−0.02 (−0.05, 0.00)	0.00 (−0.02, 0.03)
	Adjusted ^1^	Ref	−0.01 (−0.03, 0.02)	−0.02 (−0.05, 0.00)	0.00 (−0.02, 0.03)
Apo B/Apo A	Unadjusted	Ref	0.01 (−0.02, 0.03)	−0.01 (−0.04, 0.01)	0.02 (−0.01, 0.04)
	Adjusted ^1^	Ref	0.01 (−0.02, 0.03)	−0.02 (−0.04, 0.01)	0.01 (−0.01, 0.04)
HDL (mmol/l)	Unadjusted	Ref	−0.01 (−0.04, 0.03)	−0.02 (−0.06, 0.01)	−0.04 (−0.08, −0.00) *
	Adjusted ^1^	Ref	−0.01 (−0.04, 0.03)	−0.02 (−0.06, 0.02)	−0.03 (−0.07, 0.00)
LDL (mmol/l)	Unadjusted	Ref	−0.05 (−0.16, 0.05)	−0.18 (−0.29, −0.07) **	0.00 (−0.11, 0.11)
	Adjusted ^1^	Ref	−0.05 (−0.15, 0.06)	−0.17 (−0.28, −0.06) **	0.01 (−0.10, 0.12)
LDL/HDL	Unadjusted	Ref	−0.03 (−0.14, 0.08)	−0.13 (−0.25, −0.01) *	0.08 (−0.04, 0.20)
	Adjusted ^1^	Ref	−0.02 (−0.14, 0.09)	−0.12 (−0.25, −0.00) *	0.08 (−0.04, 0.20)
Cholesterol (mmol/l)	Unadjusted	Ref	−0.06 (−0.17, 0.05)	−0.14 (−0.25, −0.02) *	−0.05 (−0.16, 0.07)
	Adjusted ^1^	Ref	−0.05 (−0.16, 0.06)	−0.13 (−0.25, −0.02) *	−0.03 (−0.15, 0.08)
Triglycerides ^2^ (mmol/l)	Unadjusted	Ref	−0.00 (−0.06, 0.05)	−0.00 (−0.06, 0.06)	0.02 (−0.04, 0.08)
	Adjusted ^1^	Ref	−0.00 (−0.06, 0.05)	−0.01 (−0.07, 0.05)	0.01 (−0.04, 0.07)

* *P* < 0.05, ** *P* < 0.01. ^1^ Models adjusted for maternal education, SES and age. ß and 95% of CI obtained using general linear models. ^2^ The level of biomarker was transformed using the natural logarithm, and effect estimate reported accordingly. DV, Domestic violence; HDL, High Density Lipoprotein; LDL, Low Density Lipoprotein; Ref, reference category.

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
