# Peer review of "Maternal Experience of Domestic Violence, Associations with Children’s Lipid Biomarkers at 10 Years: Findings from MINIMat Study in Rural Bangladesh"

_nutrients, 2019, doi:10.3390/nu11040910_

Round 1
Reviewer 1 Report
With great interest I read your article. The hypothesis is interesting, and I think it is a logical mechanism; the relation between a disturbed lipid profile because of stress and domestic violence.
I do however, have some questions.
- I wondered why the authors send this manuscript to the journal Nutrients, since the manuscript is not about nutrients or food. It is more about Psychoneuroendocrinology.
- If it is a conscious choice, maybe the authors should add information about food intake, since the study was imbedded in an intervention trial controlling food and micronutrient intake.
- No information is available about the first 10 years of the children; are the parents divorced, are the children subject to violence (only violence to the mother), food habits, fasting times. All can affect the lipidprofile. Therefor I think the conclusion of the manuscript is not very solid, since the reader is not informed about many aspects.
- There was only 1 blood sample taken. Lipid profiles do change over time in children, even in a few months and are also influenced by food intake. When the authors can provide more samples over time, the effect can be more clear since the blood values can show a pattern instead of a single value
- please write the full word for HDL and LDL etc. in the heading Outcome variables
-The discussion was very nice and good to understand. Maybe the authors can write a possible mechanism why only Apo A is affected in any DV and in other forms of DV more the lipid profile. This can be interesting
Author Response
With great interest I read your article. The hypothesis is interesting, and I think it is a logical mechanism; the relation between a disturbed lipid profile because of stress and domestic violence.
I do however, have some questions.
- I wondered why the authors send this manuscript to the journal Nutrients, since the manuscript is not about nutrients or food. It is more about Psychoneuroendocrinology.
- If it is a conscious choice, maybe the authors should add information about food intake, since the study was imbedded in an intervention trial controlling food and micronutrient intake.
Thank you so much for spending your valuable time on the manuscript and your comments. Since biomarkers are very much related to the metabolic diseases and obesity later on in life we found the journal appropriate for the manuscript.
- No information is available about the first 10 years of the children; are the parents divorced, are the children subject to violence (only violence to the mother), food habits, fasting times. All can affect the lipidprofile. Therefor I think the conclusion of the manuscript is not very solid, since the reader is not informed about many aspects.
The main purpose of this study was to evaluate the association between maternal exposure to DV and child cardio metabolic risk factors. While all factors that you have mentioned are important, they may be on the casual pathway through which DV can affect child biomarkers. Factors on a causal pathway should not be adjusted for since this would lead to an underestimation of the effect. Thus we have not adjusted for them. However we adjusted for other important confounders such as SES and maternal education which are not seemed to be in the causal pathway.
- There was only 1 blood sample taken. Lipid profiles do change over time in children, even in a few months and are also influenced by food intake. When the authors can provide more samples over time, the effect can be clearer since the blood values can show a pattern instead of a single value
Thank you so much for your comment. As you mentioned the biomarkers might change within the time in an individual, however since this was a population based study and we have had relatively large sample size fluctuation of biomarkers in the individuals has been covered, thus one blood sample is enough. Further we have a sample of children over one full year period, which enabled us to include the seasonal changes in diet within our study population.
- please write the full word for HDL and LDL etc. in the heading Outcome variables
The changes have been made based on your suggestion and the full name were added to the text and tables.
-The discussion was very nice and good to understand. Maybe the authors can write a possible mechanism why only Apo A is affected in any DV and in other forms of DV more the lipid profile. This can be interesting.
We are afraid that we are not able to explain this, possibly the Apo A is more sensitive and thus respond faster than the other biomarkers.
Reviewer 2 Report
The article is based on consequences of maternal experience of Domestic Violence (DV) on their children's 14 cardio-metabolic risk factors of the lipid profile (lipid biomarkers).
The study has the approval of the ethical committee (# 2000-025, # 12-022 respectively and # 2012-346).
In material and methods, many variables are described that are not used later, it does not make sense to indicate them.
Line 175: indicate the units: years
Table 1: include BMI of children, indicate units (height, weight ...).
Table 4: multiple comparisons are presented, it would have to be adjusted by the number of comparisons.
Table 5. Why is it not adjusted for nutrition? It is an important confusion factor and the authors have the information.
The discussion and conclusions should be adjusted to the results, especially the lack of adjustment to nutritional factors is an important weakness in terms of assessing the lipid profile.
Author Response
- In material and methods, many variables are described that are not used later, it does not make sense to indicate them.
Thank you so much for your valuable time and comments. We have tried to explain all the variables that have been used in the models in order to give a clearer picture to the readers. Hope that is OK with you.
- Line 175: indicate the units: years
Changes have been done based on your suggestion.
- Table 1: include BMI of children, indicate units (height, weight ...).
Changes have been done based on your suggestion.
- Table 4: multiple comparisons are presented, it would have to be adjusted by the number of comparisons.
The comparisons that have been done were only between unadjusted model and adjusted models for each biomarker separately and not between biomarkers, thus no needed for adjustment for multiple comparison. Further adjustment for multiple comparison would have increased type II error in our findings (Rothman KJ, 1990). In order to reduce confusion, we have removed the line regarding multiple in the statistical section.
- Table 5. Why is it not adjusted for nutrition? It is an important confusion factor and the authors have the information.
The discussion and conclusions should be adjusted to the results, especially the lack of adjustment to nutritional factors is an important weakness in terms of assessing the lipid profile.
Thank you so much for your valuable comment. The main purpose of this study was to evaluate association between maternal exposure to DV and child cardio metabolic biomarkers. Previously we have shown that maternal exposure to DV is negatively associated with child nutritional status, that means that factors such as diet and nutrition might play a role in causal the pathway through which maternal exposure to DV can affect child biomarkers. Adjusting for such factors would have diluted the association and caused us underestimate the strength of the association.
Round 2
Reviewer 1 Report
Dear authors,
Thank you for your explanations to my questions.
I still think the conclusions are not based on strong scientific evidence; domestic violence and 10 years later 1 lipid profile. No other influences on lipid profile like diet are taken into account. Since this was a dietary intervention, there must be some differences. I would like another parameter, a second lipid profile, a stress questionnaire or something elso to support their findings and conclusions. I still think another experiment should be added to make the evidence stronger, or to describe it in the discussion as a serious limitation.
Author Response
Dear Reviewer,
We would like to thank you again for your valuable comments on our paper. Considering your concerns regarding the interventions, we have carefully checked the effects of supplementation in the models to make sure that the results have not been affected. This has been explained in the limitation. We also added some explanation regarding the lack of information on the dietary pattern of the children in the limitation part. We hope that this is OK with you.
Many thanks again.
Reviewer 2 Report
After reviewing the new manuscript of the original article "Maternal experience of domestic violence, associations with children's lipid biomarkers at 10 years: findings from MINIMat study in rural Bangladesh" (nutrients-473700).
This work addresses as a goal the consequences of maternal experience of Domestic Violence (DV) on their children's cardio-metabolic risk factors.
This study is in the MINIMat trial included a cohort of 1167 mothers and their children.
Their results suggest that maternal experience of physical or sexual DV might negatively affect their children's lipid profile at the age of 10 years.
The authors have incorporated the suggestions made, so the article I think is clearer.
Author Response
Thank you so much for your valuable time.